# Sorghum Contribution to Increased Income and Welfare of Dryland Farmer Households in Wonogiri, Indonesia

Dewi Sahara [1,*], Joko Triastono [1], Raden Heru Praptana [2], Anggi Sahru Romdon [1], Forita Dyah Arianti [3], Sugeng Widodo [1], Arlyna Budi Pustika [2], Heni Purwaningsih [4], Andi Yulyani Fadwiwati [1], Sutardi [2], Muslimin [1], Agus Supriyo [2], Afrizal Malik [1], Tota Suhendrata [5], Cahyati Setiani [1], Teguh Prasetyo [3], Komalawati [6], Munir Eti Wulanjari [1], Chanifah [1] and Endah Nurwahyuni [2]

[1]  Research Center for Behavioral and Circular Economics, National Research and Innovation Agency, Jl. Jend. Gatot Subroto No. 10, Jakarta 12710, Indonesia; joko040@brin.go.id (J.T.); angg044@brin.go.id (A.S.R.); suge018@brin.go.id (S.W.); andi062@brin.go.id (A.Y.F.); musl010@brin.go.id (M.); afri010@brin.go.id (A.M.); cahy023@brin.go.id (C.S.); muni003@brin.go.id (M.E.W.); chan007@brin.go.id (C.)

[2]  Research Center for Food Crops, National Research and Innovation Agency, Jl. Raya Bogor-Jakarta, Cibinong Bogor 16911, Indonesia; rade045@brin.go.id (R.H.P.); arlyna.budi.pustika@brin.go.id (A.B.P.); suta016@brin.go.id (S.); agus199@brin.go.id (A.S.); enda062@brin.go.id (E.N.)

[3]  Research Center for Sustainable Production System and Life Cycle Assessment, National Research and Innovation Agency, Serpong, South Tangerang City 15314, Indonesia; fori001@brin.go.id (F.D.A.); tegu035@brin.go.id (T.P.)

[4]  Research Center for Food Technology and Processing, National Research and Innovation Agency, Jl. Jogja-Wonosari KM 31.5 Gading, Playen, Gunungkidul, Yogyakarta 55861, Indonesia; heni007@brin.go.id

[5]  Research Center for Macroeconomics and Finance, National Research and Innovation Agency, Jl. Jend. Gatot Subroto No. 10, Jakarta 12710, Indonesia; tota001@brin.go.id

[6]  Research Center for Cooperatives, Corporation, and People's Economy, National Research and Innovation Agency, Jl. Jend. Gatot Subroto No. 10, Jakarta 12710, Indonesia; koma007@brin.go.id

*   Correspondence: dewi055@brin.go.id

**Abstract:** Sorghum is uniquely adapted to dryland and used by the Indonesian government to optimize the utilization of dryland and increase farmers' incomes. The objective of this study was to analyze the contribution of sorghum to increasing income and the level of welfare of dryland farmer households in Wonogiri, Central Java, Indonesia. The study was conducted from October to December 2022 using a survey method through direct interviews with purposively selected sorghum-growing farmers. A set of questions included household income and expenditure. The average total income and expenditure indicators, poverty line figures, RMW, and ERFHI were used to measure the welfare level of farmer households. The study found that the income derived from sorghum contributes 22.87% to total household income and tends to increase household income by 29.65%. According to the average total income earned, farming households were in a prosperous condition. This can be seen from the total income, which is higher than the total expenditure, the average per capita income higher than the poverty line, the average total income higher than RMW, and the value of ERFHI at 1.25. Income derived from sorghum has increased by 21.43–56.00% from less prosperous households to prosperous ones. The results of this study are expected to contribute to the existing literature on sorghum development and farmers' income and to be a reference for policy makers in formulating poverty alleviation programs and improving the welfare of farmer households, as well as expanding sorghum development by optimizing the utilization of agroecological, economic and social resources.

**Keywords:** sorghum; dryland; poverty; welfare; farmer households

## 1. Introduction

The agricultural sector, a fundamental source of human sustenance, significantly contributes to societal needs by providing food and various other products [1]. The

agricultural sector provides many economic, social, and environmental benefits, with the primary objective of agricultural sector development being to secure food availability [2]. In addition, the agricultural sector is a primary source of income for rural communities, an employment generator, and a catalyst for economic growth, capable of enhancing farmers' income and significantly contributing to the improvement of societal welfare and poverty alleviation [3]. Fluctuations in the earnings of farmers and agricultural laborers can notably influence the growth trajectory of the agricultural economy, a vital facet of Indonesia's economic development [4].

Indonesia harbors substantial dryland resources that can be utilized optimally to support the agricultural sector [5]. The utilization of dryland aligns with the imperative to expand the scale of land and elevate the cropping index, a cornerstone of future agricultural development policies globally [6]. Dryland plays a pivotal role in addressing food needs, and productive dryland can increase regional and national income and food security. Rational utilization of superior land resources and enhanced land-use efficiency can ensure both food security and environmental sustainability [7]. Indonesia boasts a dryland expanse of approximately 63.4 million hectares, with around 8.8 million hectares dedicated to dryland agriculture [8]. However, dryland development is challenged by several factors, such as limited water resources, soil fertility, organic matter content, and climate variations, thereby resulting in relatively low dryland productivity [9]. If not managed efficiently, dryland productivity could decrease significantly leading to negative impacts on food security, a reduction in income, and an increased vulnerability to poverty.

Alleviating poverty among smallholders is a primary concern in addressing climate change. Therefore, policies encouraging environmentally friendly and sustainable land productivity are necessary [10]. Dryland management, through innovative technology application and efficient water resources utilization, can enhance land productivity, mitigate the risk of soil degradation, maintain land fertility, increase farmers' income, and maintain sustainable food security [11]. The Indonesian government has implemented diverse agricultural programs, including technological innovations, to increase farmers' income and food crop production. For the past decade, one priority program has been the optimization of dryland use for producing drought-tolerant food crops, particularly sorghum [12]. Compared to other food crops, sorghum displays superior drought tolerance, exhibits broad adaptability across diverse land types, and is suitable for development in dryland or lowland wetland agroecosystems [13,14].

Wonogiri is a district in Central Java, Indonesia, characterized by dryland and rainfed land, which has been partly utilized for the development of sorghum. Local farmers in Wonogiri cultivate sorghum on rainfed land and dryland without intensive maintenance due to its easy cultivation and low input requirements [15]. Sorghum farming in Wonogiri yields considerable benefits, with farmers' perceptions of its development being highly favorable [16]. Sorghum, a preferred crop for rainfed land, greatly impacts farmers' livelihoods in areas such as Maharashtra, India [17] and Kerio Valley, Kenya [18]. It is also an agricultural commodity that displays increased output, provides employment, increases household income, and contributes to poverty reduction in Ghana, West Africa [19].

Farmers, in addition to being producers, are consumers of both agricultural and non-agricultural products. Hence, they optimize their land resources to generate income to cater to household and production needs. Agroecological potential, business size, and market access significantly interact with household income, with business size being a determinant of a household's capacity to surpass the living income threshold [20]. Giller et al. [20] described contrasting differences between food security and farm household income across various farming systems. A significant and sustainable increase in farmers' income requires a paradigm shift in the entire approach to the agricultural sector [21]. To enhance farmers' welfare, income augmentation policies through various supportive aspects of both the agricultural and non-agricultural sectors are needed [22]. Analysis of the food security status of farmers by combining household typologies and their livelihoods offers a beneficial approach to achieving poverty alleviation development targets [23]. Information

on the impact of sorghum development on elevating farmers' income and welfare based on farmers' socioeconomic characteristics and resource potential in the Wonogiri region remains insufficient. This study aims to analyze the contribution of sorghum to increasing income and welfare levels of dryland farming households in the Wonogiri Regency. The novelty of this research is the increase in the welfare of farmer households who grow sorghum based on household income and expenditure as well as regional per capita income, regional minimum wages, and exchange rates for farmer household income. The results of this study were expected to become future research literature and a reference for policy formulation in poverty alleviation programs and improving farmers' welfare through the broader development of sorghum.

## 2. Materials and Methods

### 2.1. Description of Study Areas

The study was conducted in the districts of Pracimantoro and Wuryantoro in Wonogiri Regency, Central Java, Indonesia, from October to December 2022. According to the Schmidt and Ferguson climate classification, Wonogiri has three distinct climate types, specifically, a moderately wet climate (C3) and a dry climate (D3 and D4), with an average temperature range of 24 °C to 32 °C. Wonogiri has a dryland potential of 88,868 hectares, or 48.70% of the area of Wonogiri. The primary food crops cultivated in Wonogiri include rice, maize, soybeans, peanuts, green beans, cassava, sweet potatoes, and sorghum. Wonogiri is the largest sorghum development center in Central Java, and by 2022, the area of sorghum cultivation in Wonogiri was 51.42 hectares [24]. In 2022, the total population of Wonogiri was 1,071,080, with 23.19% of this population engaged in farming activities and 9.82% classified as living in poverty [25]. The employment rate in the agricultural sector in Wonogiri increased by 14.47% from August 2021 to 2022 [26].

### 2.2. Sampling Methods

The study locations, the Pracimantoro and Wuryantoro Districts, were selected purposively because the agricultural lands in these two districts were dominated by dryland and rainfed and being the center of sorghum development center in Wonogiri. A purposive random sampling method was used to select households that employed a rice, maize, and sorghum planting pattern within a year. In 2022, the total number of farmers in Wonogiri who planted sorghum, adhering to a cropping pattern of rice, corn, and sorghum within a year was 104 people. Therefore, a total of 70 respondents, constituting 67.31% of the total, met the criteria for inclusion in the survey research [27].

### 2.3. Data Collections

Primary data collection was conducted using a survey method using direct interviews with a structured questionnaire. The questionnaire contains two main topics: household income and expenditure. Household income was categorized into on-farm income (derived from farming rice, corn, and sorghum), off-farm income (derived from farm labor wages), and non-farm income (derived from salaries of state civil servants, traders, and entrepreneurs). Household expenditure was divided into food expenditures (which include both purchased and non-purchased food items), non-food expenditures (encompassing bathing and washing supplies, educational expenditure, healthcare cost, fuel, LPG, telecommunications, electricity, water, land and building taxes, and social expenditure, such as celebrations, funerals, and monthly fees), and farming costs (including the cost of production facilities and labor wages). Before the interview, the enumerators were trained. Primary data were gathered from a single point over one year and were averaged per month. Secondary data utilized in this study included the poverty line figure and the regional minimum wage (RMW) of Wonogiri as applicable in 2022.

*2.4. Data Analysis*

2.4.1. Characteristics of Respondents

Respondent characteristics were analyzed descriptively. These characteristics were presented in tabular format, including the relevant criteria, range, average value, and corresponding percentage.

2.4.2. Source of Farmer Household Income

Household income includes all income gained from rice, corn, and sorghum farming (on-farm), farm labor (off-farm), and income from non-farm activities. The total income of farmer households was calculated mathematically using a modified sum model from the formula proposed by Hartoyo et al. [28]. The formula is as follows:

$$THI = HI_{\text{on-farm}} + HI_{\text{off-farm}} + HI_{\text{non-farm}}$$

Information:

THI (Total Household Income)      =household income (USD month$^{-1}$)
HI (Household Income)      =income from on-farm, off-farm and non-farm (USD month$^{-1}$)

2.4.3. Farmer Household Expenditure

Farmer household expenditure includes all costs incurred to fulfill household necessities, namely food needs, non-food items, and farming costs. The calculation for household expenditure, following a modification from the model used by Hartoyo et al. [28]. The formula is as follows:

$$THE = FE + NFE + PCF + S$$

Information:

THE (Total Household Expenditure)      =farm household expenditure (USD month$^{-1}$)
FE (Food Expenditure)      =food expenditure (USD month$^{-1}$)
NFP (Nonfood Expenditure)      =non-food expenditure (USD month$^{-1}$)
PCF (Production Cost of Farming)      =production costs (USD month$^{-1}$)
S (Saving)      =unspent income (USD month$^{-1}$)

2.4.4. Farmer Household Welfare Level

The welfare level of the farmer household is determined through four indicators, namely (i) the ratio between total household income and expenditure; (ii) comparison between per capita income and regional poverty line; (iii) comparison between total income and RMW; and (iv) exchange rate for farmers' household income (ERFHI). Farm household income and expenditure data obtained from interviews were used to calculate the values of these four indicators. The criteria for a household to be classified as prosperous are as follows:

i    Household income exceeds the household expenditure [29,30].
ii    The per capita income surpasses the regional poverty line [31,32]. The poverty line represents the minimum amount required by an adult to meet basic daily needs, which include food, non-food items, education, and health [33]. In 2022, the poverty line in the Wonogiri Regency was USD 24.18 per capita$^{-1}$ month$^{-1}$ [34].
iii    Household income exceeds the RMW [25]. In 2022, the RMW value in Wonogiri was USD 118.04 month$^{-1}$ [34].
iv    The exchange rate of farmers' household income (ERFHI) is the trade-off between agricultural products and products and services consumed by households [35,36]. The ERFHI value reflects the balance between total household income (THI) and total household expenditure (THE), which is formulated as follows:

$$ERFHI = {THI}/{THE}$$

Farm household welfare level: ERFHI > 1 = prosperous household; ERFHI = 1 = the welfare remains unchanged; and ERFHI < 1 = the household is not prosperous.

## 3. Results

### 3.1. Characteristic of Respondents

The majority of respondents were within the age range of 51–60 years (42.86%), followed by respondents older than 60 years (25.71%), and only 7.14% were under 40 years old. The education level of respondents was mostly between 10 and 12 years (45.71%), followed by those who had 7–9 years of education (25.71%), with a mere 5.71% having more than 13 years of education. Approximately 68.57% of respondents had less than 10 years of experience in sorghum farming, and only 10.00% of respondents had more than 20 years of experience in growing sorghum. The highest percentage of the respondents' family dependents fell within the range of 3–5 people (75.71%), with only 10.00% having more than 6 dependents. The occupations of the respondents include farmers, traders, entrepreneurs, and government employees, with 90.00% of respondents working as farmers.

On average, the respondents owned 0.6 hectares of land, and 31.43% of respondents had a land area of 0.25–1.00 hectares and 12.86% of respondents owned more than 1.0 hectares of land. Respondents had a varying range of monthly income. The majority of respondents had income ranging from USD 308.08 to USD 462.12 month$^{-1}$ (34.29%), while 15.71% of respondents had an income less than or equal to USD 115.53 month$^{-1}$. Respondents allocated income for various types of household needs, both food and nonfood. The total monthly household expenditure of respondents varied between less than or equal to USD 115.53 and more than USD 462.12 month$^{-1}$. The majority of respondents had expenditures in the range of USD 308.08–USD 462.12 month$^{-1}$ (32.86%), with 12.86% of respondents allocating income to meet household needs exceeding USD 462.12 month$^{-1}$. The characteristics of the dryland farming households in Wonogiri are presented in detail in Table 1.

**Table 1.** Characteristics of dryland farmer households in Wonogiri Regency, Central Java, Indonesia, 2022.

| Description | Criteria | Amount | Average |
|---|---|---|---|
| Age (Year) | <40 | 5 (7.14) * | 54.73 |
| | 41–50 | 17 (24.29) | |
| | 51–60 | 30 (42.86) | |
| | >60 | 18 (25.71) | |
| Education (Year) | <6 | <6 (22.86) | 10.00 |
| | 7–9 | 18 (25.71) | |
| | 10–12 | 32 (45.71) | |
| | >13 | 4 (5.71) | |
| Sorghum Growing Experience (Year) | <10 | 48 (68.57) | 10.64 |
| | 11–20 | 15 (21.43) | |
| | 21–30 | 3 (4.29) | |
| | >30 | 4 (5.71) | |
| Number of Family Dependents (Persons) | <2 | 10 (14.29) | 4.01 |
| | 3–5 | 53 (75.71) | |
| | >6 | 7 (10.00) | |
| Occupation | Farmer | 63 (90.00) | |
| | Trader | 1 (1.43) | |
| | Self-employed | 3 (4.29) | |
| | Government Employees | 3 (4.29) | |
| Area of Land Ownership (ha) | <0.25 | 22 (31.43) | 0.60 |
| | 0.26–0.50 | 19 (27.14) | |
| | 0.51–0.75 | 8 (11.43) | |
| | 0.76–1.00 | 12 (17.14) | |
| | >1.00 | 9 (12.86) | |

**Table 1.** *Cont.*

| Description | Criteria | Amount | Average |
|---|---|---|---|
| Income (USD month$^{-1}$) | ≤115.53 | 11 (15.71) | 413.90 |
| | 115.53–192.55 | 6 (8.57) | |
| | 192.55–308.08 | 6 (8.57) | |
| | 308.08–462.12 | 24 (34.29) | |
| | >462.12 | 23 (32.86) | |
| Expenditure (USD month$^{-1}$) | ≤115.53 | 12 (17.14) | 330.65 |
| | 115.53–192.55 | 11 (15.71) | |
| | 192.55–308.08 | 15 (21.43) | |
| | 308.08–462.12 | 23 (32.86) | |
| | >462.12 | 9 (12.86) | |

Noted: Primary data, 2022 (processed); * Numbers in parentheses represent percentages; 1 USD = 15,580.43 IDR, exchange rate November 2022 [37].

*3.2. Farmer Household Income*

Income from on-farm activities dominates household income, accounting for 82.74% of total income. Sorghum farming contributes 22.87% to the total household income, impacting an increase in household income by 29.65% (from USD 319.25 to USD 413.90). Income from sorghum represents the second largest source of income after income from rice farming. The smallest proportion of income is derived from farmers' activities as farm laborers, contributing 3.45% of the total income, while income from non-farm activities contributes 13.81%. The average income of farmer households from on-farm, off-farm, and non-farm activities is presented in Table 2.

**Table 2.** Sources of household income for dryland farmers in Wonogiri Regency, Central Java, Indonesia, 2022.

| Source of Income | Value (USD Month$^{-1}$) | Percentage (%) |
|---|---|---|
| On-farm: | | |
| a. Sorghum | 94.65 | 22.87 |
| b. Rice | 172.91 | 41.78 |
| c. Maize | 74.91 | 18.10 |
| Off-farm | 14.29 | 3.45 |
| Non-farm | 57.14 | 13.81 |
| | 413.90 | 100.00 |

Noted: Primary data, 2022 (processed); 1 USD = 15,580.43 IDR, exchange rate November 2022 [37].

*3.3. Farmer Household Expenditure*

The largest proportion of household expenditure to meet non-food needs is USD 113.90 month$^{-1}$, while spending on food is only USD 82.29 month$^{-1}$, lower than non-food expenditure. In addition to food and non-food needs, farmers also allocate costs for farming activities, averaging USD 134.46 month$^{-1}$ constituting 40.67% of total expenditure. The largest proportion of household expenditure to meet food and non-food needs is 47.40% of total income, while 32.49% of income is used for production costs. High production costs are allocated for planting food crops three times a year. After subtracting total expenditure, farmers still have unspent income of USD 83.25 month$^{-1}$ or 20.11% of total income. Farm household expenditures were presented in detail in Table 3.

**Table 3.** Types of dryland farmer household expenditure in Wonogiri Regency, Central Java, Indonesia, 2022.

| Types of Expenditure | Value (USD Month$^{-1}$) | Percentage to (%) | |
| --- | --- | --- | --- |
| | | Expenditure | Income |
| Food | | | |
| a.     Non-purchased Food | 19.29 | 5.84 | 4.66 |
| b.     Purchased Food | 62.99 | 19.05 | 15.22 |
| Non-food | 113.90 | 34.45 | 27.52 |
| Production Cost | 134.46 | 40.67 | 32.49 |
| Total Expenditure Amount | 330.64 | 100.00 | |
| Residual Income | 83.25 | | 20.11 |
| Total Number | 413.90 | | 100.00 |

Noted: Primary data, 2022 (processed); 1 USD = 15,580.43 IDR, exchange rate November 2022 [37].

### *3.4. Farmer Household Welfare*

3.4.1. Household Income and Expenditure

The average income of farmer households was USD 413.90 month$^{-1}$ (Table 2), which is higher than the average expenditure of USD 330.65 month$^{-1}$ (Table 3). This suggests that farmer households in the study locations are in a prosperous condition. However, based on the income of individual households, not all respondent farmer households are in a prosperous condition; 24.29% of the farmer households are not prosperous (Table 4).

**Table 4.** Income and expenditure of dryland farmers' households in Wonogiri Regency, Central Java, Indonesia, 2022.

| Description | Income (USD Month$^{-1}$) | Expenditure (USD Month$^{-1}$) | Number of Households |
| --- | --- | --- | --- |
| Income < Expenditure | 67.22–279.62 | 104.46–332.60 | 17 (24.29) * |
| Income > Expenditure | 155.00–1140.43 | 110.63–976.42 | 53 (75.71) * |

Noted: Primary data, 2022 (processed); * Numbers in parentheses represent percentages; 1 USD = 15,580.43 IDR, exchange rate November 2022 [37].

3.4.2. Per Capita Income Is Greater than the Poverty Line

The average per capita household income is USD 118.19 per capita$^{-1}$ month$^{-1}$ which is significantly higher than the Wonogiri poverty line of USD 24.18 per capita$^{-1}$ month$^{-1}$. The transition from non-prosperous households can be observed based on the average per capita income per month of each individual household, both with and without sorghum, as detailed in Table 5. Household income without sorghum ranges from USD 5.71–USD 205.98 per capita$^{-1}$ month$^{-1}$, individually, and 14 households (20.00%) fall below the poverty line, with incomes ranging from USD 8.56 to USD 18.61 per capita$^{-1}$ month$^{-1}$, while 56 households (80.00%) exceed the poverty line, with incomes between USD 25.05 and USD 205.98 per capita$^{-1}$ month$^{-1}$. In contrast, households with income from sorghum have a range of USD 8.56 to USD 262.29 per capita$^{-1}$ month$^{-1}$. In this case, there are 9 households (12.86%) below the poverty line and 61 households (87.14%) above the poverty line. Thus, sorghum farming contributes to increasing per capita monthly income for households and has made five additional households prosperous (Table 5).

**Table 5.** Household welfare based on per capita income in Wonogiri Regency, Central Java, Indonesia, 2022.

| Description | Household Income without Sorghum (USD per Capita$^{-1}$ Month$^{-1}$) | Number of Households | Total Household Income (USD per Capita$^{-1}$ Month$^{-1}$) | Number of Households | The Regional Poverty Line of Wonogiri Regency 2022 (USD per Capita$^{-1}$ Month$^{-1}$) |
|---|---|---|---|---|---|
| Household Income Below the Poverty Line | 5.71–24.18 | 14 (20.00) * | 8.56–18.61 | 9 (12.86) * | |
| Household Income Above the Poverty Line | 25.05–205.98 | 56 (80.00) * | 25.35–262.29 | 61 (87.14) * | 24.18 ** |

Noted: Primary data, 2022 (processed); * Numbers in parentheses represent percentages; ** Source: Statistic of Wonogiri [24]; 1 USD = 15,580.43 IDR, exchange rate November 2022 [37].

3.4.3. Income Greater than Regional Minimum Wage (RMW)

Farmer households are in a prosperous condition with an average total income of USD 413.90, which exceeds the RMW of USD 118.04. Table 6 illustrates that the increase in sorghum farming can reduce the number of households under poverty, based on each household's total income. Individually, 20.00% of households fall below the RMW without sorghum, with income ranging from USD 51.35 to USD 114.57. Meanwhile, 80.00% of households rise above the RMW with incomes between USD 118.20 and USD 785.17. When considering household income with sorghum, the number of households living below the RMW drops to 15.71% with a total income range of USD 67.22 to USD 85.58. Concurrently, 84.29% of households rise above the RMW with a total income ranging from USD 161.70–USD 903.31 (Table 6).

**Table 6.** Household welfare based on the regional minimum wage (RMW) indicator, in Wonogiri Regency, Central Java, Indonesia, 2022.

| Description | Household Income without Sorghum (USD Month$^{-1}$) | Number of Households | Total Household Income (USD Month$^{-1}$) | Number of Households | The RMW of Wonogiri Regency 2022 (USD Month$^{-1}$) |
|---|---|---|---|---|---|
| Household Income Below the RMW | 51.35–114.57 | 14 (20.00) * | 67.22–85.58 | 11 (15.71) * | |
| Household Income Above the RMW | 118.20–785.17 | 56 (80.00) * | 161.70–903.31 | 59 (84.29) * | 118.04 ** |

Noted: Primary data, 2022 (processed); * Numbers in parentheses represent percentages; ** Source: Statistic of Wonogiri [24]; 1 USD = 15,580.43 IDR, exchange rate November 2022 [37].

3.4.4. Exchange Rate of Farmer Household Income (ERFHI)

The ERFHI value obtained in this study was 1.25. As per the welfare criteria, household welfare is achieved if the ERFHI value is greater than 1. Consequently, households in the study locations, in aggregate, are considered prosperous. Table 7 demonstrates that sorghum farming contributes to an increase in the income of each household and reduces the number of households that are not yet prosperous. If the ERFHI value is calculated based on the income of each household without the sorghum income, then 25 households (35.71%) would not be considered prosperous. However, if the ERFHI value is determined from the total household income including the sorghum income, the number of households that are not yet prosperous decreases to 11 households (15.71%) (Table 7).

**Table 7.** Household welfare based on exchange rate of farmer household income (ERFHI) in Wonogiri Regency, Central Java, Indonesia, 2022.

| Description | Household Income without Sorghum (USD Month$^{-1}$) | Number of Households | Total Household Income (USD Month$^{-1}$) | Number of Households |
|---|---|---|---|---|
| ERFHI < 1 | 51.35–114.57 | 25 (35.71) * | 67.22–85.58 | 11 (15.71) * |
| ERFHI > 1 | 118.20–785.17 | 45 (84.29) * | 161.70–903.31 | 59 (84.29) * |

Noted: Primary data, 2022 (processed); * Numbers in parentheses represent percentages; 1 USD = 15,580.43 IDR, exchange rate November 2022 [37].

## 4. Discussion

### 4.1. Characteristics of Respondents

Sorghum farming in Wonogiri is predominantly conducted by farmers aged over 50 years. Age plays a role in influencing the technical operations in the farming process and consequently, the income generated. Age can also impact farmers' understanding of agricultural technology and innovation [38]. Younger farmers tend to be more receptive to the utilization of modern technology and novel methods to enhance farming productivity and efficiency [39], whereas older farmers typically operate with traditional methods, hesitant to adopt new technologies due to their reliance and trust in conventional methods [40]. Nevertheless, the level of education amongst respondents was generally satisfactory, with an average of 10 years of education. This positively influenced their understanding of sorghum cultivation techniques and management. Higher levels of education can improve farmers' knowledge of advanced cultivation techniques, efficient land management methods and sustainable agricultural practices [41,42], and farmers' ability to comprehend and implement better marketing strategies [43]. Lower levels of education and poor physical health are more prevalent among impoverished farmers with low incomes and living standards [44]. The level of education can also affect farmers' access to information related to sorghum. Farmers with higher education levels are better equipped to identify, understand, and utilize the information necessary to enhance production, quality, and farming efficiently and sell their products at profitable prices.

While the primary occupation of most respondents was farming, those who worked as traders, entrepreneurs, and government employees also made optimal use of their land for sorghum cultivation, akin to farmers. The Wonogiri farmers' experience in cultivating sorghum has spanned an average of 10 years. Farming experience is a factor that significantly influences farmer participation in sorghum cultivation [45], and high farmer participation has positively impacted the adoption of sustainable sorghum cultivation [46]. Prolonged experience in sorghum farming can lead to deeper knowledge and skills in sorghum management as well as to the identification of best cultivation practices. Sorghum cultivation contributes to the sustainability of farmer households in Wonogiri. This reflects the high participation of farmers in cultivating sorghum as a source of farming income. The contribution of sorghum to the income and welfare of farmer households is highly reliant on the success of the production process. Farmers who prioritize sorghum farming tend to have superior skills and knowledge, which affects the quantity and quality of sorghum production as well as the income earned.

The average number of dependents in a respondent's family was four people. The number of family dependents influences the success of farming and income [47]. Income and the number of family dependents affect the welfare of the farmer's household [48]. A large number of dependents can affect the allocation of resources and income derived from farming. Farmer households with many dependents tend to have higher needs, which encourages them to increase production from farming. A large number of family dependents can also affect the allocation of resources and time that must be arranged for farming and fulfilling family responsibilities. The number of family members actively involved and providing support in farming can positively impact the performance, income, and welfare of farmer households.

Most of the farmers own more than 0.25 hectares of land, and they plant sorghum on rainfed land after rice or corn during the dry season and on dryland. Because sorghum can thrive both on dryland and previously unproductive land, farmers can obtain crop yields even under unfavorable weather conditions [49,50]. Farmers in Wonogiri believe that sorghum is an apt commodity to be planted on degraded land, and in addition to being easy to manage, sorghum also serves as a source of supplemental income from farming. Wonogiri farmers' perceptions of the ease of obtaining sorghum seeds, seed growth, plant growth, plant maintenance, resistance to pests and diseases, sorghum production, sorghum markets, sorghum marketing, processing of sorghum seeds, sorghum prices, and profits from sorghum farming are quite favorable [16]. The area of land ownership can affect the contribution of sorghum to income. Farmers with larger landholding areas have the potential to earn more income from planting sorghum and contribute to an increase in their total income and welfare [51]. Moreover, broader land ownership also affords opportunities for diversification of farming and integrated farming systems.

Respondents have sources of income from on-farm, off-farm, and non-farm activities. The agricultural sector was the primary source of income for most of the respondents, especially from farming food crops, namely rice, corn, and sorghum (on-farm). Another source of income is farm laborers (off-farm), while non-farm income is derived from employee wages/salaries, the advantages of self-employment profits, and family transfers. Higher incomes allow farmers to invest more in resources to increase farm production, such as high quality seeds, fertilizers, and other supporting materials. Sorghum cultivation provides farmers an opportunity to earn additional income from farming diversification [52]. In India, transitioning from rice to sorghum cultivation can enhance farmers' income [53]. Most of the income earned by Wonogiri farmers was allocated to provide farming production inputs, followed by expenditures for food and non-food needs for household sustainability. Farmers who can adeptly manage finances have controlled expenditures, prioritizing outlays needed for farming investments and mitigating unwanted financial risks. Properly managed expenditures can ensure that the income derived from farming can be used to meet daily needs, education, health, and overall household economic development.

### 4.2. Farmer Household Income

Household income from various sources depends on the geographical, economic, and social conditions of a region. Household income consists of all cash receipts, goods, or services received by households or individual family members [54]. Farmers' sources of income in Central Java and Yogyakarta, Indonesia are primarily from agricultural cultivation, livestock farming, and a small portion from non-farm businesses, such as construction work and trading [55]. Farmers in Ghana generate income from multiple sources, including rice and other commodity farming, as well as non-farm activities [56]. Farmers' participation in both single- and multi-commodity markets exhibits a positive correlation with household income. Therefore, diversification of income sources influences the increasing total income and household welfare [57]. Households with diverse income sources generally have relatively sustainable livelihoods [58].

The income from sorghum farming remains below that from rice farming, as sorghum has not been managed intensively and farmers' experience in rice farming exceeds that of sorghum farming. Wonogiri farmers began cultivating sorghum in the 1980s using traditional methods, whereas rice cultivation which commenced broadly in the 1960s, had already adopted advanced cultivation technologies such as the use utilization of superior varieties, organic fertilizers, and manure. However, sorghum farming contributes to increasing income, which is closely tied to the level of household welfare. Sorghum farming can diversify farmers' sources of income [59] as well as contribute to the enhancement of income and welfare of farmer households [60]. Farmer households' income in Ubon Ratchathani, Thailand primarily comes from agriculture, especially rice crops, followed by income from additional work (off-farm) [61]. Farmers rely heavily on agricultural businesses as their main source of income, although non-farm activities also play an

important role in earning income [62]. Zeeshan et al. [63] suggested that by engaging in non-farm businesses, farmers could yield positive gains in agricultural income and consumption expenditure for rural farming households.

Income derived from non-farm activities contributes less than a quarter of the total income of the dryland farmer household in Wonogiri. In several countries, non-farm income constitutes a significant source of income and accounts for almost half of the total income of smallholder households [64]. Non-farm income sources contribute about a third of total income. Non-farm income also contributes to increasing income inequality and significantly aids in the reduction of poverty in farming households in Kedah [65]. Farmers' income in sub-Saharan Africa comes from non-farm activities between 30 and 40% of total revenue [66]. In Kosovo, a quarter of household income comes from non-farm income and has a positive impact on poverty alleviation [67]. Rakotoarisoa and Kaitibie [18] emphasized that rural households should create opportunities to engage in non-farm activities to generate additional income as a reliable asset. Income from the non-farm sector can be reinvested as operational costs for agricultural businesses to increase productivity and household welfare [68].

### *4.3. Farmer Household Expenditure*

Household expenditure is allocated to fulfill food needs (purchased and non-purchased food) and non-food and production costs. Food expenditures include the purchase of staple foodstuffs other than rice. Non-food expenditure encompasses education, health, transportation, communication, and social expenditure. Production costs include the cost for seeds, fertilizers, pesticides, supporting facilities, and labor wages. Rubhara et al. [69] mention the criteria for farmer household expenditures, including food, production inputs, education, health, and durable goods. Farmer household expenditures depend on the returns from farming, which are used to meet the needs of daily life [70].

Farmer household expenditure for non-food items is greater than for food. As much as 47.40% of the total income was used to meet food and non-food needs, which is slightly higher than farming costs. Food is a basic necessity that must be met before other needs, while non-food needs for daily operational requirements are secondary necessities that must be met first for household sustainability. Sutrisma et al. [71] posited that the higher the level of income, the more expenditure will shift from spending on food to non-food items. Households with a lower percentage of spending on food than on non-food are considered wealthy households [72]. Wealthier households spend less income on consumption needs, while poorer households tend to allocate income to meet food needs [73]. A study by Susanti et al. [74] revealed that the biggest expenditure of farming households in Aceh Besar and Central Lampung, Indonesia, is allocated for food needs, amounting to 63.7% and 80.94%, respectively. Farmers with high expenditure continue to allocate income to meet the basic needs of household members and have not yet obtained wide access to non-food needs [75]. Connor et al. [55] stated that if farmers' income increases, their expenditure will also increase to purchase higher quality food and production inputs. An increase in farmers' income raises the ability of farmers to buy non-subsidized fertilizers [76] and provides more access to better education and health facilities [77]. Income that is not spent serves as an investment or as a buffer against potential shock to farming income [78] or unforeseen incidents [79].

### *4.4. Farmer Household Welfare Level*

4.4.1. Household Income and Expenditure

Income derived from on-farm activities greatly contributes to the welfare of the farmer households. The contribution of on-farm income from sorghum ranks second after rice. Wulandari [80] asserts that the level of income contribution to a farmer's household welfare depends on the amount of income farmers received from farming activities, and farming income is influenced by productivity. Increased agricultural productivity encourages greater investment among small farmers and can enhance household welfare and livelihoods [81].

Increased productivity affects enhancing household income, thus leading to higher purchasing power, which in turn increases expenditure for daily household needs [82]. The contribution of income from sorghum can be augmented by increasing productivity through the optimization of geographical, economic, and social resources. Widodo et al. [16] reported that sorghum productivity in Wonogiri is still low, ranging from 2.4 to 3.0 t ha$^{-1}$. The productivity of sorghum on dryland can reach 5.24–5.95 t ha$^{-1}$ [83,84], indicating that sorghum productivity in Wonogiri has the potential for improvement. Farming income is calculated as revenue (output multiplied by price) minus the total cost of farming [85]. While farmers cannot control both input and output prices, they can control productivity, making it a determinant factor for farm income. Assuming that the price of output is fixed, an increase in sorghum production will be followed by an increase in income. Furthermore, Widodo et al. [16] suggested that an increasing income could be realized through sales in the form of processed products.

The proportion of expenditure on food is often used as a benchmark for household welfare [86–88]. The allocation of household income to food consumption forms a significant portion of overall expenditure. Household consumption behavior provides a more realistic portrayal of farmers' welfare levels [75]. A household is categorized as prosperous if it can meet the needs of all family members. This condition can be achieved if households have a higher total income than their total expenditure.

### 4.4.2. Per Capita Income Was Greater than the Poverty Line

Total household income and the number of family dependents determine the per capita income of the farming households. Sorghum farming contributes to an increase in household income, resulting in an aggregate average per capita income of farming households in Wonogiri that is considerably above the poverty line. Individually, the number of households considered unprosperous decreased from 14 to 9, reflecting an increase in welfare of 35.71%. Onuche and Oladipo [32], however, presented different results, indicating that as many as 61.87% of farmer households in Kogi State, Central Nigeria, fall below the poverty line based on per capita income due to limited resources and suboptimal productivity. Meanwhile, farmer households in Wonogiri, gain income from on-farm and off-farm activities, as well as non-farm activities. Poverty in agricultural areas tends to be higher if farmers lack non-farm sources of income [89]. Henry et al. [90] noted that apart from income factors, the poverty status of farmer households in the Central Zone of Plateau State, Nigeria is influenced by the education level, age, number of family dependents, and farming experience.

### 4.4.3. Income Was Greater than Regional Minimum Wage (RMW)

The RMW value is commonly used as a standard for the welfare level of farmer households in Indonesia. Farmers' income from cultivating food crops and horticulture, and other incomes in Muara Sabak Timur, Indonesia, significantly exceeds the RMW, indicating prosperity in these farming households [91]. Sorghum farming in Wonogiri has contributed to an increase in household welfare based on the minimum wage value. This improvement is demonstrated by a decrease from 14 to 11 in the number of unprosperous households, signifying an increase of 21.43% unprosperous to prosperous households. These results suggest that not only sorghum farming is a profitable commodity that contributes to increased income, but it also has the potential to prevent farmers from migrating to other areas in pursuit of additional income. Farmers tend to optimize their resources to increase farming productivity to obtain a higher income. Irawan and Yuristia [29] stated that if agricultural conditions in rural areas are not utilized to their full potential, the existence of greater source of income in urban areas encourages farmers to migrate for additional income.

### 4.4.4. Exchange Rate of Farmer Household Income (ERFHI)

The ERFHI value of 1.25 indicated that implementing sorghum farming has been proven to improve the welfare of farmer households. The income derived from upland

rice farming in Boyolali and lowland rice in Gorontalo, Indonesia also contributes to the improvement of farmer households' welfare, as evidenced by the respective ERFHI values of 1.36 [28] and 2.75 [36]. The increasing income from sorghum farming has improved the status of 56.00% of farmer households from unprosperous to prosperous. However, generally, the raising income from sorghum in farmer households with smaller landholdings and greater family dependents has not managed to offset expenditures to meet food and non-food needs. Thus, it is important to manage resources from the geographical, economic, and social dimensions of the region to increase the productivity of sorghum farming and opportunities for other sources to achieve the welfare of the farmer household as a whole.

Sargani et al. [10] suggest the need for a strategic dryland management policy to increase farmer income and food security. The findings of this study can serve as a reference for policymakers in formulating poverty alleviation programs and improving farmer households' welfare through the expansion of the sorghum farming business. These results also support the study of Widodo et al. [16] concerning the strategy for developing sorghum in Wonogiri, based on the potential strength of regional resources, economic value, and farmers' perceptions. This strategy is outlined in the operational policies, including improving productivity through optimizing sorghum cultivation techniques in accordance with agroclimatic conditions, and cooperating with off takers to guarantee the price, increase income, and as an incentive to boost production and sustainable processing of products. Nonetheless, this study has limitation in that the parameters used to measure welfare level were solely based on total income and expenditure compared to the poverty line, RMW, and ERFHI. Therefore, more comprehensive studies that integrate the structure of income and expenditure with determinants of food security, poverty and farmer household welfare are necessary in the future.

## 5. Conclusions

Sorghum is a drought-tolerant food crop which is a priority commodity of the Indonesian government in its dryland optimization program to increase farmers' income. Wonogiri has dryland potential of 88,868 hectares and is one of the target areas for sorghum development. Sorghum contributed to an increase in the income of Wonogiri dryland farmers by 22.87% of total household income and impacted an increase in income of 29.65%. Based on the average of total income earned, farmer households are in a prosperous condition, with total income (USD 413.90 month$^{-1}$) greater than total expenditure (USD 330.65 month$^{-1}$), the average income per capita higher than the poverty line figure (USD 24.18 per capita$^{-1}$ month$^{-1}$), the average total income greater than that the regional minimum wage (RMW = USD 118.04 month$^{-1}$), and the exchange rate of farmer household income (ERFHI) is 1.25. Increased income from sorghum contributed to raising 21.43–56.00% of farmer households from unprosperous to prosperous. The contribution of sorghum to the income and welfare of farmer households could be enhanced by increasing the productivity and added value of sorghum through optimization of the use of agroecological, economic, and social resources of the region. The results of this study are expected to be used as a reference for policymakers in formulating poverty alleviation programs and improving the welfare of farmer households through the development of a broader sorghum farming business. The implication of this study are innovations to improve dryland management through the implementation of environmentally friendly and sustainable agricultural practices, efficient use of water resources, and collaborative approaches between the government, sorghum farmers, and the private sector. In achieving a balanced link between dryland management, food security and farmers' income, strategic policies through incentives and technical assistance to dryland farmers are needed. The novelty of this study is the use of four indicators of the welfare level of farmer households that have demonstrated an increase in farmer households' income and welfare based on an analysis of total household income and expenditure. Further comprehensive studies need

to be carried out that integrate the income and expenditure structures with determinants of food security, poverty, and farmer household welfare.

**Author Contributions:** Conceptualization, D.S., J.T., R.H.P., A.S.R., F.D.A., S.W., A.B.P., H.P., A.Y.F., S., M., A.S., A.M., T.S., C.S., T.P., K., M.E.W., C. and E.N.; methodology, D.S., J.T., A.S.R., F.D.A., S.W., M., A.M., T.S., C.S., T.P., K., M.E.W. and C.; software, R.H.P., A.S.R., S., A.S., K., M.E.W. and C.; validation, D.S., J.T, R.H.P., S.W., A.B.P., H.P., A.Y.F., A.S., A.M., T.S., C.S., T.P. and E.N.; formal analysis, D.S., J.T., R.H.P., A.S.R., F.D.A., A.B.P., H.P., A.Y.F., S., M., A.S., A.M., T.S., C.S., T.P., K., M.E.W., C. and E.N.; investigation, D.S., J.T., R.H.P., A.S.R., F.D.A., S.W., A.B.P., H.P., A.Y.F., S., M., A.M., T.S., C.S. and T.P.; resources, D.S., J.T., R.H.P., A.S.R., F.D.A., S.W., A.B.P., H.P., A.Y.F., S. and M.E.W.; data curation, D.S., J.T., R.H.P., A.S.R., F.D.A., A.B.P., H.P., A.Y.F., S., M., A.M., T.S., C.S., T.P. and M.E.W.; writing—original draft preparation, D.S., J.T., R.H.P., A.S.R., F.D.A., S.W., A.B.P., H.P., A.Y.F., S., M., A.S., A.M., T.S., C.S., T.P., K., M.E.W., C. and E.N.; writing—review and editing, D.S., J.T., R.H.P., A.S.R., F.D.A., S.W., A.B.P., H.P., A.Y.F., S., M., A.S., A.M., T.S., C.S., T.P., K., M.E.W., C. and E.N.; visualization, D.S., J.T., R.H.P., A.S.R., F.D.A., S.W., A.B.P., H.P., A.Y.F., S., M. and A.S.; funding acquisition, S.W. All authors have read and agreed to the published version of the manuscript.

**Funding:** This research was funded by the National Research and Innovation Agency of Indonesia through Research Organization for Governance, Economy, and Community Welfare through research project entitled Prospects and Potential for Sorghum Development to Support Self-Reliance. in Food, Animal Feed and Energy in the Central Area of Yogyakarta and Central Java. Grant number. B-1832/III.12/PR.03.08/9/2022.

**Institutional Review Board Statement:** Not applicable.

**Data Availability Statement:** The data presented in this study are available upon request from the corresponding author. The data are not publicly available yet but will be in due course.

**Acknowledgments:** The authors express their gratitude to the National Research and Innovation Agency of Indonesia, Department of Agriculture and Food of Wonogiri Regency, and thanks to Sugeng Hariyadi, who helped in collecting data in the field.

**Conflicts of Interest:** The authors declare no conflict of interest.

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
