# Peer review of "Sorghum Contribution to Increased Income and Welfare of Dryland Farmer Households in Wonogiri, Indonesia"

_agriculture, doi:10.3390/agriculture13081609_

Round 1
Reviewer 1 Report
The Abstract should be reworked. The author should concentrate on presenting the methods and summarizing the findings. No need to indicate particular figure in USD. Instead, the author should focus on discussing the revealed tendencies and trends. Also, the implications of the findings should be emphasized. In addition to that, the author should briefly outline potential contribution of the study to the literature and policies.
In the Introduction, the author should elaborate the linkages between food security, farmers' income, and drylands. This triangle is not demonstrated adequately in the Introduction, but it is definitely the core issue of the study. How does food security depend on drylands, what are the challenges? Similarly, what is the income-drylands relationship, how do they affect each other? The generalization of these issues is needed, the overall relevance of the study should be discussed first prior to proceeding with the country-specific relevance of the study for Indonesia. The discussion of the gaps in previous and current studies should be expanded sufficiently to address the security-income-drylands triangle. Specific gaps should be revealed, and the aim of the study must be articulated respectively.
Lines 110-111 - why 70 people? The author should provide details here - how were respondents selected, what was the approach, etc.
Lines 115-116 - the author should provide a questionnaire sample. What kind of interviews - face-to-face, group, online, etc.? A thorough description of the method is absolutely needed here.
Line 128 - how was the impact of sorghum farming on the total income measured, if total income comes from several sources?
Line 155 - why these particular parameters? The critical discussion of variables along with their selection are needed. It is not clear what particular questions are included in the questionnaire and why these particular questions?
In the Conclusion, the author should demonstrate whether the aim of the study has been achieved or not. Major findings must be summarized, and the novelty of the study should be demonstrated. Currently, I fail to see how this study is novel in any of its elements (methodology, results, contributions, implications, anything). To my mind, the potential contribution of the study to the literature is marginal. No contribution to the methodology of surveying, while the very approach to the survey is unclear and uncertain
The quality of the English language and style must be improved substantially
Author Response
Dear Reviewer 1,
Here are our responses to questions and suggestions for improvements to our article:
The Abstract should be reworked. The author should concentrate on presenting the methods and summarizing the findings. No need to indicate particular figure in USD. Instead, the author should focus on discussing the revealed tendencies and trends. Also, the implications of the findings should be emphasized. In addition to that, the author should briefly outline potential contribution of the study to the literature and policies.
- Thank you. We've reworked the abstract section as suggested. We've summarized the background study, clarified the methodology further, omitted the USD figures, indicated trending findings, and added the implications of the findings. A revised abstract was presented in the manuscript at lines 24-41.
In the Introduction, the author should elaborate the linkages between food security, farmers' income, and drylands. This triangle is not demonstrated adequately in the Introduction, but it is definitely the core issue of the study. How does food security depend on drylands, what are the challenges? Similarly, what is the income-drylands relationship, how do they affect each other? The generalization of these issues is needed, the overall relevance of the study should be discussed first prior to proceeding with the country-specific relevance of the study for Indonesia. The discussion of the gaps in previous and current studies should be expanded sufficiently to address the security-income-drylands triangle. Specific gaps should be revealed, and the aim of the study must be articulated respectively.
- Thank you. We have improved the introductory section regarding the elaboration of the relationship between food security, farmer income and dryland supported by some appropriate literature. These corrections were presented in the manuscript at lines 56-74.
Lines 110-111 - why 70 people? The author should provide details here - how were respondents selected, what was the approach, etc.
- Thank you. We have added an explanation regarding the sample size of respondents. Sampling of respondents was done purposively. In 2022, there were 104 farmers who growth sorghum with the rice-corn-sorghum cropping pattern. The number of respondents as many as 70 people (67.31%) fulfilled the criteria for the number of respondents for survey research supported by related literature. Improved explanations are presented in the manuscript at lines 132-136.
Lines 115-116 - the author should provide a questionnaire sample. What kind of interviews - face-to-face, group, online, etc.? A thorough description of the method is absolutely needed here.
- Thank you. We have rewritten the data collection section by adding an explanation of the survey method used, the topic set of questions in the questionnaire, and the data obtained from the survey activity. These corrections were presented in the manuscript at lines 139-152.
Line 128 - how was the impact of sorghum farming on the total income measured, if total income comes from several sources?
- Thank you. We have added information regarding the revenue contribution and impact of increased income from sorghum on total revenue. Revenue contribution from sorghum was the percentage of income from sorghum to total income, while the impact of income from sorghum on increasing income was the percentage increase in income from sorghum to total income other than sorghum. These corrections were presented in the manuscript at lines 242-243.
Line 155 - why these particular parameters? The critical discussion of variables along with their selection are needed. It is not clear what particular questions are included in the questionnaire and why these particular questions?
- Thank you. We have improved the explanation regarding the four indicators to measure the welfare level of farmer households. To get the value of each of these indicators, data on household income and expenditure was needed. Household income and expenditure variables were described in detail in the topic set of questions in the questionnaire as presented in the revised data collection section of the manuscript at lines 139-152. Welfare indicators No. 1 and 2 have been used by several studies in a number of countries, while indicators No. 3 and 4 have been commonly used in Indonesia. In this study we used these four indicators for the case of sorghum in dryland farmers. Improved explanations of the four indicators were presented in the manuscript at lines 189-195.
In the Conclusion, the author should demonstrate whether the aim of the study has been achieved or not. Major findings must be summarized, and the novelty of the study should be demonstrated. Currently, I fail to see how this study is novel in any of its elements (methodology, results, contributions, implications, anything). To my mind, the potential contribution of the study to the literature is marginal. No contribution to the methodology of surveying, while the very approach to the survey is unclear and uncertain.
- Thank you. We have rewritten the conclusions indicating that our research objectives have been achieved through the methodology, data obtained, indicators used and analysis that we carried out. We consider that the study of the contribution of sorghum to increasing the income and welfare of Wonogiri dryland farmers was a novelty considering that the sorghum development program in Wonogiri will only start in 2021. The agroecological conditions of the Wonogiri dryland may have similarities with dryland in other regions, but the social and economic characteristics of farmers varies between regions. The cropping pattern practiced by farmers from previously rice-corn-fallen to rice-corn-sorghum was one of the potential new backgrounds for developing policy implications for increasing income through optimizing dryland and diversifying on-farm income. The approach of using the four indicators to measure the welfare level of a farmer's household has never been done before. We have added the benefits and policy implications of the research findings. Improved conclusions were presented in the manuscript on line 568-596.
Thank you for your consideration of this manuscript.
Sincerely,
Anggi Sahru Romdon

Reviewer 2 Report
This paper explores the contribution of sorghum to increasing income and welfare levels of dryland farmer households. The topic is of great significance. This paper use a survey method to find that sorghum contributes to an increase in the income of farmers by 22.87% of total household income. The paper is relatively clear. However, the article needs further improvement in the following aspects.
1. The structure of paper needs to be adjusted. The paper is suggested to adjust into the following parts: the first part, introduction and literature review. The second part, research program, including method and data. The third part is empirical analysis. The fourth part is further analysis. The fifth part is the conclusion.
2. The introduction needs to be revised. The introduction gives too much background and requires only a concise paragraph. Besides that, in the introduction part, it is better to explain the research objective of this paper and the interesting conclusion obtained in this paper, as well as the innovation or contribution of this paper, so that readers can understand this paper more easily.
3. This paper lacks a literature review. Literature review is a summary of previous studies, which can clearly reflect the current research status and highlight the innovation or contribution of this paper. Therefore, it is suggested that the author add a literature review Part. For example, agriculture are related to this topic, so it needs to be further supplemented. Therefore, it is suggested that the author refer to the following literatures and systematically reviewed them.
[1] Jiehua Ma, Shuanglian Chen. Does land transfer have an impact on land use efficiency? A case study on rural China[J]. National Accounting Review, 2022, 4(2): 112-134. doi: 10.3934/NAR.2022007
[2] Jinhui Zhu, Mengxin Wang, Changhong Zhang. Impact of high-standard basic farmland construction policies on agricultural eco-efficiency: Case of China[J]. National Accounting Review, 2022, 4(2): 147-166. doi: 10.3934/NAR.2022009
[3] Li Z., Chen H. & Mo B. (2022), Can digital finance promote urban innovation? Evidence from China, Borsa Istanbul Review, https://doi.org/10.1016/j.bir.2022. 10.006.
4. It is suggested to summarize the contribution of the article and placed it in the Introduction part.
5. It is recommended to add theoretical analysis before building the model so as to explain why the methods used in this paper are suitable for the objectives of this paper.
6. The economic explanation of the empirical results is insufficient. In this paper, each empirical result is only to explain whether the relevant test statistics are significant or not, but the explanation of economics is not enough. It is suggested that the graph and table involving the empirical results should be combined for economic explanation. In fact, statistical significance can only explain the "what", but does not explain the "why", that is, the paper needs to explain the "why" based on the econometric analysis.
7. The empirical analysis in this paper is insufficient to draw this conclusion. It is suggested that the author use more specific data and conduct more rigorous empirical analysis to verify the conclusion.
8. Please read the whole text and correct the grammar, spelling and other mistakes.
Moderate editing of English language required.
Author Response
Dear Reviewer, 2.
Here are our responses to questions and suggestions for improvements to our article:
Comments and Suggestions for Authors
This paper explores the contribution of sorghum to increasing income and welfare levels of dryland farmer households. The topic is of great significance. This paper use a survey method to find that sorghum contributes to an increase in the income of farmers by 22.87% of total household income. The paper is relatively clear. However, the article needs further improvement in the following aspects.
- Thank you for appreciating our article.
- The structure of paper needs to be adjusted. The paper is suggested to adjust into the following parts: the first part, introduction and literature review. The second part, research program, including method and data. The third part is empirical analysis. The fourth part is further analysis. The fifth part is the conclusion.
- Thank you. We have accommodated the suggested article structure. However, we followed the article template provided in this journal issue in setting out the article structure.
- The introduction needs to be revised. The introduction gives too much background and requires only a concise paragraph. Besides that, in the introduction part, it is better to explain the research objective of this paper and the interesting conclusion obtained in this paper, as well as the innovation or contribution of this paper, so that readers can understand this paper more easily.
- Thank you. Still related to suggestion No. 1, we have improved the introductory section by sharpening research objectives, innovation and contribution of research results.
- This paper lacks a literature review. Literature review is a summary of previous studies, which can clearly reflect the current research status and highlight the innovation or contribution of this paper. Therefore, it is suggested that the author add a literature review Part. For example, agriculture are related to this topic, so it needs to be further supplemented. Therefore, it is suggested that the author refer to the following literatures and systematically reviewed them.
[1] Jiehua Ma, Shuanglian Chen. Does land transfer have an impact on land use efficiency? A case study on rural China[J]. National Accounting Review, 2022, 4(2): 112-134. doi: 10.3934/NAR.2022007
[2] Jinhui Zhu, Mengxin Wang, Changhong Zhang. Impact of high-standard basic farmland construction policies on agricultural eco-efficiency: Case of China[J]. National Accounting Review, 2022, 4(2): 147-166. doi: 10.3934/NAR.2022009
[3] Li Z., Chen H. & Mo B. (2022), Can digital finance promote urban innovation? Evidence from China, Borsa Istanbul Review, https://doi.org/10.1016/j.bir.2022. 10.006.
- Thank you. We have improved the introduction by adding two literature No. 1 and 2. These improvements were presented in the manuscript at line 56-61. We have read literature No. 3, sorry we did not add it in the introduction because it is not related to the topic of this study. We will consider the literature to support our next article.
- It is suggested to summarize the contribution of the article and placed it in the Introduction part.
- Thank you. We have summarized and placed the contributions of this study in the introduction to the manuscript in the lines 108-111.
- It is recommended to add theoretical analysis before building the model so as to explain why the methods used in this paper are suitable for the objectives of this paper.
- Thank you. In our view, the research used the theory of welfare with four indicators (income and expenditure, poverty line, RMW and ERFHI) which were analyzed based on income and expenditure data of farmer households.
- The economic explanation of the empirical results is insufficient. In this paper, each empirical result is only to explain whether the relevant test statistics are significant or not, but the explanation of economics is not enough. It is suggested that the graph and table involving the empirical results should be combined for economic explanation. In fact, statistical significance can only explain the "what", but does not explain the "why", that is, the paper needs to explain the "why" based on the econometric analysis.
- Thank you. This study has used econometric analysis but has not been able to explain the significance between variables because this study identified the contribution of income from sorghum to increasing the income and welfare of farmer households on dryland. The study did not identify determinants of the welfare of farmer households.
- We have put the findings in table form to answer "what", while to explain "why" we put them in discussion. For example: 1) The results of the study showed that sorghum contributes to the income of dryland farmers in Wonogiri by 22.87% of total household income and has an impact on increasing income by 29.65% to answer "what" was presented in the manuscript on line 242-243, while the answer to "why" was presented in the manuscript on line 423-429; 2) The largest proportion of household expenditure to meet food and non-food needs was 47.40% of total income to answer “what” was presented in the manuscript on line 258-259, while to answer “why” was presented in the manuscript on line 463-465.
- The empirical analysis in this paper is insufficient to draw this conclusion. It is suggested that the author use more specific data and conduct more rigorous empirical analysis to verify the conclusion.
- Thank you. The empirical data on household income and expenditure analyzed with the four indicators of welfare have answered the research objectives that could draw conclusions supported by descriptive qualitative discussion of the results of the analysis.
- Please read the whole text and correct the grammar, spelling and other mistakes.
- Thank you. We have read the whole text and corrected the grammar, spelling and other mistakes.
Thank you for your consideration of this manuscript.
Sincerely,
Anggi Sahru Romdon

Reviewer 3 Report
Recommendations:
1. Strengthen the literature review to contextualize the study within existing research.
2. Provide a more detailed methodology, including sample size, data collection methods, and potential biases.
3. Conduct a comprehensive analysis of factors contributing to increased income from sorghum cultivation.
4. Contextualize and compare the welfare levels with other regions or time periods.
5. Discuss the study's limitations to better understand its scope and potential biases.
6. Offer actionable recommendations and policy implications based on the findings.
7. Suggest potential areas for future research to expand on the study's impact and implications.
8. Differentiation of Off-farm and non-farm income sources.
9. Can the authors explain why income and expenditures are measured in USD rather than Indonesian currency?
10. Limited scope: The study focuses solely on the contribution of sorghum to increased income and welfare, neglecting potential negative impacts or trade-offs associated with sorghum cultivation.
11. Lack of control group: The study lacks a control group or comparative analysis with farmers who do not cultivate sorghum, making it challenging to isolate the specific impact of sorghum on income and welfare.
12. Generalizability: The study's findings may not easily apply to other regions or countries with different agroecological conditions and socioeconomic contexts.
13. Reliance on self-reported data: The study relies on farmers' self-reported income and expenditure data, which could be subject to recall bias or inaccuracies.
14. Inadequate data on expenses: The study does not provide detailed information on farmers' expenses, making it difficult to understand the full economic implications of sorghum cultivation.
15. Absence of long-term analysis: The study's limited timeframe (October to December 2022) may not capture potential seasonal variations in sorghum income and the sustainability of its impact.
16. Lack of qualitative data: The study primarily focuses on quantitative indicators, missing valuable insights that could be gained from qualitative data, such as farmers' perceptions and experiences with sorghum cultivation.
17. Ignoring non-economic factors: The study overlooks non-economic factors, such as social dynamics and cultural aspects, which can significantly influence farmers' well-being and decisions.
18. Limited policy implications: The study does not offer comprehensive policy recommendations or strategies to support the sustainable adoption of sorghum cultivation.
19. Overemphasis on average data: Relying solely on average data may hide disparities among farmer households and fail to capture the challenges that specific subgroups face within the community.
20. The introduction and discussion section can be improved and strengthened by comparing the following studies.
https://doi.org/10.1007/s10668-022-02296-5
https://doi.org/10.1016/j.jrurstud.2023.103035
7. Conclusion:
The conclusion reiterates the findings but fails to comprehensively summarize the study's implications for policy and practice. Additionally, it does not suggest future research directions or offer recommendations for maximizing sorghum's contribution to the welfare of dryland farmers.
8. Overall Assessment:
The study highlights sorghum's potential to increase dryland farmer households' income and welfare in Wonogiri, Indonesia. However, several areas need improvement to strengthen the report's credibility and impact. The study lacks a robust literature review, detailed methodology, comprehensive analysis of contributing factors, and a discussion of limitations. Moreover, the conclusion must be more comprehensive, providing actionable insights for policymakers and stakeholders. Finally, the inclusion of many authors in a study seems to be obvious and uncommon. Can the researcher reflect on this research project's collaborative nature and complexity?
Extensive editing of English language required
Author Response
Dear Reviewer 3,
Here are our responses to questions and suggestions for improvements to our article:
- Strengthen the literature review to contextualize the study within existing research.
- Thank you. We have added some supporting literature to contextualize this study. These additional refinements were presented in the manuscript at lines 56-74, 133-136, 385-389 and 555-556.
- Provide a more detailed methodology, including sample size, data collection methods, and potential biases.
- Thank you. We have added an explanation related to the sample size of respondents. These additions were presented in the manuscript at lines 133-136. We have also included a detailed explanation regarding the method of data collection and potential bias as presented in the manuscript at lines 139-152.
- Conduct a comprehensive analysis of factors contributing to increased income from sorghum cultivation.
- Thank you. We did not analyze the factors that contributed to the increase in income from sorghum cultivation, but we only reported income from sorghum cultivation carried out by farmers in one growing season at that time. So it was assumed that the factors that contribute to the derived of sorghum, such as production input prices and selling prices, were the prevailing ones at that time. We added this explanation to the discussion section in subchapter 4.4.1. lines 497-501.
- Contextualize and compare the welfare levels with other regions or time periods.
- Thank you. We did not compared welfare levels between regions and over time. The data collected was income and expenditure data from one point in time and one rice-corn-sorghum cropping pattern in a year. The four welfare indicators used were only valid for a period of one year.
- Discuss the study's limitations to better understand its scope and potential biases.
- Thank you. We have described the drawbacks of this study in the manuscript at lines 565-567. In our opinion, there was another weakness that the research location was carried out in one area. Due to the potential for bias, we have added a data collection section to the manuscript at lines 139-152.
- Offer actionable recommendations and policy implications based on the findings.
- Thank you. We have added actionable recommendations and policy implications based on the findings in the manuscript at lines 586-592.
- Suggest potential areas for future research to expand on the study's impact and implications.
- Thank you. We have added potential areas for future research to expand the impact and implications of research in the manuscript at lines 597-599.
- Differentiation of off-farm and non-farm income sources.
- Thank you. We have added an explanation of the differentiation of off-farm and non-farm sources of income in the data collection section of the manuscript in line 139-152.
- Can the authors explain why income and expenditures are measured in USD rather than Indonesian currency?
- Thank you. We measured income and expenditure using the IDR, but because this article will be published at an international level, we changed the IDR to USD as suggested by this editorial journal in our previous article. https://doi.org/10.3390/agriculture13030516
- Limited scope: The study focuses solely on the contribution of sorghum to increased income and welfare, neglecting potential negative impacts or trade-offs associated with sorghum cultivation.
- Thank you. We agree that this research only focuses on the contribution of sorghum to increasing household income and welfare. Previously, the Wonogiri dryland farmers only applied the rice-corn-fallow cropping pattern. Meanwhile, in the last ten years, the Indonesian government launched a program to optimize the use of dryland and increase farmers' income. Sorghum was a priority commodity to be developed on dryland because sorghum is a plant that is adaptive to dryland. Wonogiri was one of the targets for the development of sorghum which was planted after corn fills land that was previously fallow. So this research focuses on the impact of sorghum's contribution to increasing household income and welfare as a policy reference for wide-scale development of sorghum.
- Lack of control group: The study lacks a control group or comparative analysis with farmers who do not cultivate sorghum, making it challenging to isolate the specific impact of sorghum on income and welfare.
- Thank you. Still related to the topic of discussion No. 10, in our study we only focused on the contribution of income and the impact of increasing income from sorghum on total income and household welfare. So that the respondent sample was selected purposively, namely farmers who growth sorghum with a rice-corn-sorghum cropping pattern. Revenue contribution from sorghum was the percentage of income from sorghum to total income, while the impact of income from sorghum on increasing income was the percentage increase in income from sorghum to total income other than sorghum.
- Generalizability: The study's findings may not easily apply to other regions or countries with different agroecological conditions and socioeconomic contexts.
- Thank you. We agree with the statement that the study's findings may not easily apply to other regions or countries with different agroecological conditions and socioeconomic contexts, because each region has specific characteristics of agroecological and socioeconomic resources. This study reviewed the contribution of sorghum to increasing income and welfare of farmers in the context of the agroecological and socioeconomic characteristics of Wonogiri dryland farmers which were expected to be a reference or comparison for cases in other dryland areas.
- Reliance on self-reported data: The study relies on farmers' self-reported income and expenditure, which could be subject to recall bias or inaccuracies.
- Thank you. We agree with this statement, that this study relies on income and expenditure data reported by farmers themselves during interviews, not based on farm report keeping. To anticipate potential bias from the data, the team has coached the enumerators before conducting interviews. We have added a description of the existence of a coaching enumerator in the data collection section of the manuscript at line 149-150.
- Inadequate data on expenses: The study does not provide detailed information on farmers' expenses, making it difficult to understand the full economic implications of sorghum cultivation.
- Thank you. We have detailed the types of farm household expenditure in the topic set of questions in the questionnaire. We have added a detailed description of farm household expenditures to the manuscript in the data collection section of line 144-149.
- Absence of long-term analysis: The study's limited timeframe (October to December 2022) may not capture potential seasonal variations in sorghum income and the sustainability of its impact.
- Thank you. This study used primary data on household income and expenditure obtained at one point in time in the rice-corn-sorghum cropping pattern in one year. While the secondary data was in the form of the poverty line and RMW which were valid for 2022. The poverty line and RMW values ​​in each district change every year so that strategic policy implications were needed from this research to be able to capture long-term changes.
- Lack of qualitative data: The study primarily focuses on quantitative indicators, missing valuable insights that could be gained from qualitative data, such as farmers' perceptions and experiences with sorghum cultivation.
- Thank you. With regard to qualitative data, we recorded information from respondents during interviews not only in addition to quantitative data but also an explanation of the data. We used the data explanation information to discuss the research results. Information on farmers' perceptions was presented in the manuscript on line 383-389, while information on farmers' experiences in cultivating sorghum was presented in the manuscript on line 423-428.
- Ignoring non-economic factors: The study overlooks non-economic factors, such as social dynamics and cultural aspects, which can significantly influence farmers' well-being and decisions.
- Thank you. The data we collect was in the form of farm household income and expenditure. Related to non-economic factors such as social dynamics and cultural aspects, we include these factors in the types of household expenditure contained in the questionnaire on the topic set of household expenditure in the non-food category. We have added this information in the manuscript data collection section at the line 144-149.
- Limited policy implications: The study does not offer comprehensive policy recommendations or strategies to support the sustainable adoption of sorghum cultivation.
- Thank you. We have added comprehensive policy or strategy recommendations to support the adoption of sustainable sorghum cultivation in the manuscript on line 586-594.
- Overemphasis on average data: Relying solely on average data may hide disparities among farmer households and fail to capture the challenges that specific subgroups face within the community.
- Thank you. To see the level of household welfare in aggregate based on average income and expenditure. However, we also analyzed the contribution of sorghum to individual household income and welfare so that the impact of its contribution can be seen.
- The introduction and discussion section can be improved and strengthened by comparing the following studies.
https://doi.org/10.1007/s10668-022-02296-5
https://doi.org/10.1016/j.jrurstud.2023.103035
- Thank you. To improve and strengthen the rationale and discussion, we have added these two literatures to the manuscript introductory section in line 69-74, and in the discussion section in line 555-556.
- Conclusion:
The conclusion reiterates the findings but fails to comprehensively summarize the study's implications for policy and practice. Additionally, it does not suggest future research directions or offer recommendations for maximizing sorghum's contribution to the welfare of dryland farmers.
- Thank you. We have summarized the main findings of this research and added policy implications and suggested future research directions to maximize sorghum for the welfare of dryland farmers. Improved conclusions were presented in the manuscript in line 571-599.
- Overall Assessment:
The study highlights sorghum's potential to increase dryland farmer households' income and welfare in Wonogiri, Indonesia. However, several areas need improvement to strengthen the report's credibility and impact. The study lacks a robust literature review, detailed methodology, comprehensive analysis of contributing factors, and a discussion of limitations. Moreover, the conclusion must be more comprehensive, providing actionable insights for policymakers and stakeholders. Finally, the inclusion of many authors in a study seems to be obvious and uncommon. Can the researcher reflect on this research project's collaborative nature and complexity?
- Thank you. This research required interpretation and broader views from various multidisciplinary perspectives, so this research project involves authors from various fields of expertise to produce comprehensive analysis and discussion. The author's areas of expertise include agricultural socio-economics, agronomy, postharvest, sustainable production, macroeconomics, and peoples economy.
Sincerely,
Anggi Sahru Romdon

Round 2
Reviewer 1 Report
The author has adequately and sufficiently addressed my Round 1 recommendations. The author should involve the native speaker of English to proofread the manuscript and ensure the best possible quality of the English language and style
The author should involve the native speaker of English to proofread the manuscript and ensure the best possible quality of the English language and style
Author Response
Dear Reviewer, 1
The author has adequately and sufficiently addressed my Round 1 recommendations. The author should involve the native speaker of English to proofread the manuscript and ensure the best possible quality of the English language and style.
The author should involve the native speaker of English to proofread the manuscript and ensure the best possible quality of the English language and style.
Thank you for your suggestions. We have improved the English Language in the manuscript as suggested, by involving the native speaker. We expect that this manuscript has already fulfilled the standard of English language required.
We hope to hear the good news from you.
Best regards,
Anggi Sahru Romdon
